# ENFORCING PHYSICAL CONSTRAINTS IN NEURAL NETWORKS THROUGH DIFFERENTIABLE PDE LAYER

## ABSTRACT

Recent studies at the intersection of physics and deep learning have illustrated successes in the application of deep neural networks to partially or fully replace costly physics simulations. Enforcing physical constraints to solutions generated by neural networks remains a challenge, yet it is essential to the accuracy and trustworthiness of such model predictions. Many systems in the physical sciences are governed by Partial Differential Equations (PDEs). Enforcing these as hard constraints, we show, are inefficient in conventional frameworks due to the high dimensionality of the generated fields. To this end, we propose the use of a novel differentiable spectral projection layer for neural networks that efficiently enforces spatial PDE constraints using spectral methods, yet is fully differentiable, allowing for its use as a layer in neural networks that supports end-to-end training. We show that its computational cost is cheaper than a regular convolution layer. We apply it to an important class of physical systems – incompressible turbulent flows, where the divergence-free PDE constraint is required. We train a 3D Conditional Generative Adversarial Network (CGAN) for turbulent flow superresolution efficiently, whilst guaranteeing the spatial PDE constraint of zero divergence. Furthermore, our empirical results show that the model produces realistic flow fields with more accurate flow statistics when trained with hard constraints imposed via the proposed novel differentiable spectral projection layer, as compared to soft constrained and unconstrained counterparts.

## 1 INTRODUCTION

Convolutional Neural Network (CNN) based deep learning architectures have achieved huge success in many tasks across computer vision, but their use in the physical sciences have only recently been explored. Many parallels exist between physical science problems and those in computer vision. For instance, grid-based simulations generate a physical scalar or vector field which can been compared to multidimensional arrays in computer vision. However, unlike computer vision problems, physical fields are often constrained by PDEs that arise from the governing equations of the physical system. For example, the Poisson equation of the form $\nabla^2 \phi = f$ is often encountered in heat diffusion problems, whereas the divergence-free (also known as solenoidal) conditions in the form of $\nabla \cdot \phi = 0$ is fundamental to magnetic fields, as well as incompressible fluid velocity fields to ensure conservation of mass. For meaningful application of deep learning to a range of important physical problems it is essential to enforce such spatial PDE constraints to guarantee physical consistency and reliability of the model output for scientific applications. Yet, general means of enforcing these constraints do not exist and the existing methods do not scale well with high dimensional, high resolution outputs.

In this paper, we address this issue by proposing a novel differentiable PDE layer (PDEL) that efficiently enforces spatial PDE constraints for neural networks, at costs on par with a single CNN layer. We use spectral methods, which leverages the highly efficient Fast Fourier Transform (FFT) algorithm for enforcing such constraints. Using this formulation, we are able to exploit the structures of the spectral matrices corresponding to these differential operators that renders the entire layer $\mathcal{O}(n \log n)$ for processing a 3 dimensional field of size $n$. The method is general for enforcing arbitrary linear combinations of differential operators on these fields, which encompasses physical constraints from a broad range of important scientific and engineering systems. We apply this hard constraining layer to the problem of turbulence superresolution, where we show that training with the

hard constraining layer in-the-loop not only guarantees that the imposed constraint is strictly satisfied, but also generates solutions that are more accurate measured via a variety of fluid flow metrics.

In summary, the main contributions of this paper are:

- We propose the highly efficient differentiable spatial linear PDE layer (PDEL), which strictly enforces linear spatial PDEs constraints.

- We apply the PDE layer towards the superresolution task for turbulent flows, showing that training with hard constraints in-the-loop results in solutions that not only strictly satisfy the imposed constraint but also produce flow fields with more accurate fluid flow statistics.

## 2    BACKGROUND AND RELATED WORK

### 2.1    CONSTRAINTS IN NEURAL NETWORKS

Many studies in machine learning have considered imposing some form of constraints for their respective applications. **?** proposed an approach for constraining the prediction of a discriminative predictor and showed that a Gaussian Process can be forced to satisfy linear and quadratic constraints. **?** proposed training a kernalized latent variable model that imposes equality and inequality constraints. **?** proposed a constrained CNN, which phrases the training objective as a biconvex optimization for linear models, which is then relaxed to nonlinear deep networks for any set of linear constraints on the output space. **?** proposed an alternative approach by randomly subsampling a set of constraints at each optimization step and projecting the gradients onto the feasible solution space. OptNet (**?**) solves a generic quadratic programming problem differentiably within the neural network, but its cubed complexity does not handle high dimensional output. **?** proposed parameterizing the feasible solution space for imposing inequality constraints. However, methods to impose physical constraints into machine learning and deep learning models remains largely unexplored.

### 2.2    APPLICATIONS OF MACHINE LEARNING FOR PDE AND TURBULENCE

More recently there have been some work in the literature to apply machine learning to PDE and physical problems. **???** developed physics informed deep learning frameworks for assimilating observational data. In the context of applying machine learning methodologies to turbulence problems, earlier works using data-driven modeling approaches have used optimization and Bayesian inference approaches to calibrate existing turbulence models (**?????**). With the advancement of efficient and accurate modeling tools in machine learning, recent studies have looked at data driven approaches for turbulence modeling (**?**), including the direct use of random forests for modeling Reynolds-Averaged Navier Stokes (RANS) model errors (**?**), the use of multilayer perceptrons for modeling Reynolds stress anisotropy tensor from simulation data (**?**), and using random forests to predict mean velocities of turbulent flow fields. **?** use a random forest to compute particle location and **?** use a CNN to approximate part of a numerical PDE scheme. A recent application of deep learning for generating realistic fluid flow fields is TempoGAN (**?**) that uses a specialized discriminator network for temporal coherence. None of these methods, however, enforce constraints that are necessary for physically consistent fluid flow fields. **?** addressed this with a customized loss function for the divergence-free constraint of fluid flow, but since it is a loss-based soft constraint, the conditions are ultimately not exactly satisfied.

## 3    PROBLEM STATEMENT

The main focus of this paper is to introduce a novel and efficient method for imposing spatial linear PDE constraints to the outputs of Convolutional Neural Networks (CNNs). This is discussed in the context of underdetermined systems, since solutions do not exist for overdetermined systems, while solutions for determined systems do not fit in the context of constraining the outputs. More specifically, given the output of the network to be a discretization of a 3D vector field $\boldsymbol{f} : \mathbb{R}^3 \mapsto \mathbb{R}^3$ and a linear spatial PDE operator $A((\frac{\partial}{\partial x_j})^0, (\frac{\partial}{\partial x_j})^1, \cdots)$ that maps vector fields to scalar fields $A\boldsymbol{f} : \mathbb{R}^3 \mapsto \mathbb{R}$, we seek a means of efficiently imposing the spatial linear PDE constraints within

CNNs, i.e.,

$$A\boldsymbol{f} = \boldsymbol{b} \tag{1}$$

Note that this form encompasses a wide range of physically relevant constraints. In particular, all spatial PDE constraints composed of divergence, curl, Laplacian and other higher order partial differential terms in linear combination may be expressed in this form. Depending on the domain of application, this includes mass conservation for incompressible fluid flows, the heat equation, the wave equation, Laplace's equation, Helmholtz equation, Klein-Gordon equation, and Poisson's equation. For the important constraint of mass conservation in incompressible flows, we investigate the divergence-free (solenoidal) constraint of:

$$\nabla \cdot \boldsymbol{f} = \sum_j \frac{\partial}{\partial x_j} \boldsymbol{f}_j = \boldsymbol{0} \tag{2}$$

## 4 METHODS

Before presenting our proposed method for enforcing the solenoidal condition on CNN outputs, we present an overview of two commonly utilized strategies for enforcing general linear constraints, which we will compare and benchmark against in the experiments section (Sec. 5).

We first discuss enforcing linear constraints on the outputs of the neural network, where we have a neural network that learns the function mapping $f : \mathbb{R}^t \mapsto \mathbb{R}^m$, where the function $f(\boldsymbol{x}; \boldsymbol{\theta})$ is parameterized by learnable parameters $\boldsymbol{\theta}$, and is subject to the linear constraint $Af(\boldsymbol{x}; \boldsymbol{\theta}) = \boldsymbol{b}$, where $A \in \mathbb{R}^{n \times m}, \boldsymbol{b} \in \mathbb{R}^n$. For this to be an underconstrained system, we have $n < m$.

### 4.1 GENERALIZED LINEAR CONSTRAINTS

Two forms of constraints are possible for explicitly enforcing a certain set of constraints for neural network outputs: soft constraints and hard constraints.

**Soft constraints** are easy to implement, by adding a differentiable residual loss for penalizing the network during training time for violating the explicit constraints. For simplicity, let $\boldsymbol{y} := f(\boldsymbol{x}; \boldsymbol{\theta})$. In the conventional unconstrained case, assume the neural network is trained under the differentiable loss function $L(f(\boldsymbol{x}; \boldsymbol{\theta}))$, in the constrained case, the loss function can be augmented by an additional residual loss term defined by:

$$L_c(\theta) = L(\theta) + \alpha((A\boldsymbol{y} - b)^T (A\boldsymbol{y} - b)) \tag{3}$$

where $\alpha$ is a hyper-parameter weighing the two loss functions that can be difficult to determine and vary between applications. Although easy to implement, soft constraints provide no guarantees on the solutions satisfying the imposed constraint.

**Hard linear constraints** can be enforced by posing the problem as a constrained optimization problem for seeking the closest point in the solution space subject to the constraints, which can be solved by satisfying the Karush-Kuhn-Tucker (KKT) condition. The result of the projection step can be written as the stationary point of the Lagrangian:

$$\min_{\hat{\boldsymbol{y}}} \max_{\boldsymbol{\lambda}} \mathcal{L}(\hat{\boldsymbol{y}}, \boldsymbol{\lambda}) \tag{4}$$

where we have the Lagrangian as:

$$\mathcal{L}(\hat{\boldsymbol{y}}, \boldsymbol{\lambda}; \boldsymbol{y}) = \frac{1}{2}(\boldsymbol{y} - \hat{\boldsymbol{y}})^T (\boldsymbol{y} - \hat{\boldsymbol{y}}) + \boldsymbol{\lambda}^T (A\boldsymbol{y} - \boldsymbol{b}) \tag{5}$$

$$\frac{\partial \mathcal{L}}{\partial \hat{\boldsymbol{y}}} = \boldsymbol{y} - \hat{\boldsymbol{y}} + \boldsymbol{\lambda}^T A \tag{6}$$

The KKT condition leads to the following linear system, the solution of which involves solving a linear system of dimensions $(m + n) \times (m + n)$. Given that the linear system is symmetric and positive definite, the solution can be sought by inverting the system:

$$\begin{bmatrix} I & A^T \\ A & 0 \end{bmatrix} \begin{bmatrix} \hat{\boldsymbol{y}} \\ \boldsymbol{\lambda} \end{bmatrix} = \begin{bmatrix} \boldsymbol{y} \\ \boldsymbol{b} \end{bmatrix} \Rightarrow \begin{bmatrix} \hat{\boldsymbol{y}} \\ \boldsymbol{\lambda} \end{bmatrix} = \begin{bmatrix} I & A^T \\ A & 0 \end{bmatrix}^{-1} \begin{bmatrix} \boldsymbol{y} \\ \boldsymbol{b} \end{bmatrix} \tag{7}$$

While this approach is general for enforcing arbitrary linear constraints on arbitrary network outputs, it is difficult to scale it to higher dimensions, and particularly difficult for 2-dimensional and 3-dimensional outputs, by direct matrix inversion followed by matrix multiplication.

## 4.2 SPECTRAL METHODS

First, we introduce and review the spectral methods (**?**) for discretizing the spatial PDE operators. Spectral methods are a class of numerical methods that computes the spatial partial derivatives of a field based on the spectral decomposition of the signal. By decomposing the original signal into a linear combination with respect to trigonometric basis functions of varying wavenumbers (or frequencies), the spatial derivatives with respect to the trigonometric basis functions can be easily and efficiently computed. The Fast Fourier Transform (FFT) is a well known algorithm for efficiently computing the Direct Fourier Transform (DFT) of uniform discrete signals. The multidimensional FFT and inverse FFT respectively compute the following:

$$\boldsymbol{F^{(k)}} = \sum_{\boldsymbol{n}=0}^{\boldsymbol{N}-1} \boldsymbol{f^{(n)}} e^{-i2\pi \boldsymbol{k}\cdot(\boldsymbol{n}/\boldsymbol{N})}; \ \boldsymbol{f^{(n)}} = \frac{1}{N_1 N_2 N_3} \sum_{\boldsymbol{k}=0}^{\boldsymbol{N}-1} \boldsymbol{F^{(k)}} e^{i2\pi \boldsymbol{k}\cdot(\boldsymbol{n}/\boldsymbol{N})} \tag{8}$$

where $\boldsymbol{F^{(k)}} = \text{FFT}(\boldsymbol{f^{(n)}}), \boldsymbol{f^{(n)}} = \text{IFFT}(\boldsymbol{F^{(k)}})$, spatial indices $\boldsymbol{n} = (n_1, n_2, n_3), n_j \in \{0, 1, \cdots, N_j - 1\}$ and spectral indices $\boldsymbol{k} = (k_1, k_2, k_3), k_j \in \{0, 1, \cdots, N_j - 1\}$. The spatial derivative with respect to $x_j$ can be computed by:

$$\frac{\partial}{\partial x_j} \boldsymbol{f^{(n)}} = \text{IFFT}(ik_j \boldsymbol{F^{(k)}}) \tag{9}$$

In matrix form, for the $t$-th component of a 3 dimensional vector field: $\boldsymbol{F}_t$, taking its flattened vector form, and taking the flattened vector form of the wavenumber $\boldsymbol{k}_j$ corresponding to the dimension $x_j$, the spatial derivative with respect to $x_j$ can be computed using matrix multiplication:

$$\frac{\partial}{\partial x_j} \boldsymbol{F}_t = diag(i\boldsymbol{k}_j) \boldsymbol{F}_t \tag{10}$$

where $diag()$ converts a vector into a corresponding diagonal matrix. In general, arbitrary linear combination of spatial derivatives of varying orders can be computed using a single diagonal matrix multiplication:

$$\left(\sum_j \sum_r c_{jr} \left(\frac{\partial}{\partial x_j}\right)^r\right) \boldsymbol{F}_t = diag\left(\sum_j \sum_r c_{jr} (i\boldsymbol{k}_j)^r\right) \boldsymbol{F}_t := A_t \boldsymbol{F}_t \tag{11}$$

where $A_t$ is a diagonal matrix for the spatial derivatives corresponding to the $t$-th component of the vector field, that is a polynomial of $i\boldsymbol{k}_j$, and $A = [A_1, A_2, A_3]$.

## 4.3 SPECTRAL PROJECTION LAYER

For brevity, we present our main results for computing the spectral projection operator that efficiently enforces spatial linear PDE constraints using spectral methods. We defer readers to Eqns (14 - 27) of Appendix A for detailed derivation of these results. In spectral space, the projection of the original vector field $\boldsymbol{F}$ into solution space: $\hat{\boldsymbol{F}}$, can be computed by:

$$\hat{\boldsymbol{F}} = P\boldsymbol{F} + Q\boldsymbol{B} \tag{12}$$

where $\boldsymbol{F} = \text{FFT}(\boldsymbol{f}), \boldsymbol{B} = \text{FFT}(\boldsymbol{b})$, and

$$P = I - \frac{1}{\sum_{j=0}^3 A_j^2} \begin{bmatrix} A_1^2 & A_1 A_2 & A_1 A_3 \\ A_1 A_2 & A_2^2 & A_2 A_3 \\ A_1 A_3 & A_2 A_3 & A_3^2 \end{bmatrix} ; Q = -\frac{1}{\sum_{j=0}^3 A_j^2} \begin{bmatrix} A_1 \\ A_2 \\ A_3 \end{bmatrix} \tag{13}$$

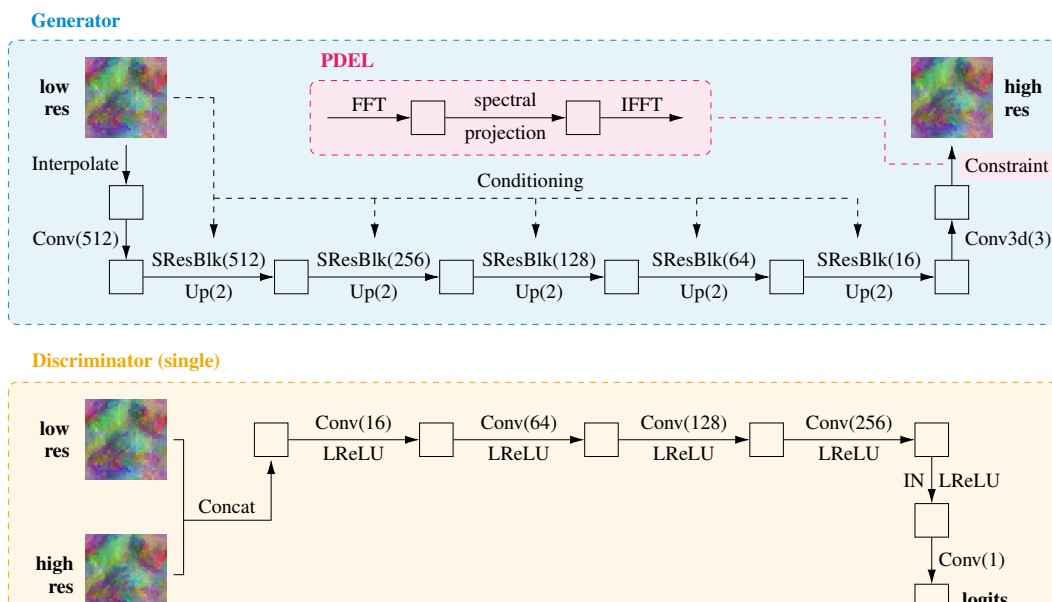

Figure 1: Schematic for the network architecture for the turbulent flow superresolution task. The network is a modified GauGAN (**?**) architecture for 3D fields that utilizes spatially-adaptive normalization for conditioning with the input, and residual blocks for facilitating gradient flows. The network inputs a low resolution flow field in $\mathbb{R}^{32 \times 32 \times 32}$ and outputs an output field of $\mathbb{R}^{128 \times 128 \times 128}$ with an upscaling factor of 4 in each dimension.

## 5 EXPERIMENTS AND RESULTS

### 5.1 COMPUTATIONAL COMPLEXITY AND COST

We first show that although the classic Lagrangian based hard constraining method in Eqn. 7 is general and able to enforce hard linear constraints, solving it by direct inversion leads to poor computational efficiency, especially with high-resolution 3-dimensional data outputs from CNNs.

Given Eqn. 7 for enforcing hard linear constraints using Lagrangian multipliers, we estimate the computational complexity regarding enforcing solenoidal conditions as follows. Without loss of generality, assume that the vector field on which we enforce the solenoidal constraints is 3 dimensional of resolution $N$ in each spatial dimension, with a total of $n = N^3$ nodes. The overall degrees of freedom in the system is $3n$, and enforcing solenoidal constraint for each voxel results in $n$ linear constraints, hence the resulting linear system in Eqn. 7 is of dimensions $4n \times 4n$. Though the matrix inversion is shared, hence reusable by caching, each projection involves a matrix multiplication of $\mathcal{O}((3n)^2) \sim \mathcal{O}(9n^2)$ operations. In comparison, the spectral projection method only involves element-wise operations, resulting in an overall complexity of $\mathcal{O}(n)$ operations for enforcing constraints, and $\mathcal{O}(n \log(n))$ for the FFT and IFFT operations. Results of an empirical analysis for computational time and memory usage is shown in Fig. 2.

### 5.2 TURBULENCE SUPERRESOLUTION WITH CONDITIONAL GAN

**Conditional Generative Adversarial Networks** While Generative Adversarial Networks (GANs) have been effective at generating 2D (**???**) and 3D (**?**) images, the unconditional generative modeling scenario of generating entire fields from random latent vectors is hardly useful in scientific settings. A more desirable model is one that is conditioned upon a set of inputs from either partial observations, or low resolution simulations due to computational limitations **?**. Conditional GANs, which have been widely used in various image-to-image translation problems (**?????**), and a recent extension of conditional GANs, the GauGAN (**?**), utilizes a novel spatially-adaptive normalization layer that better preserves semantic information in the conditional input and produces improved texture outputs.

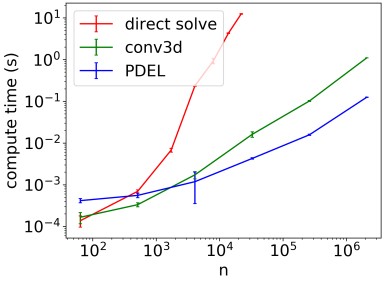
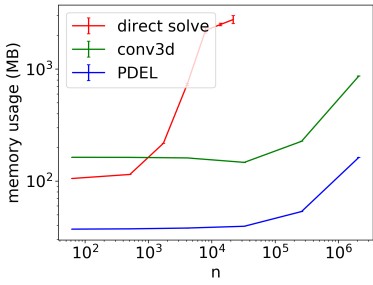

(a) Compute time vs system size        (b) Peak memory usage vs system size

Figure 2: Comparison of computational performance for direct solve using Lagrangian multiplier method and our PDE Layer (PDEL). Computational performance for a single Conv3d layer of kernel size 3 is also included for comparison purposes. Our PDEL fairs well with respect to size of the linear system, even compared with the highly optimized 3D convolution layers, allowing for its direct integration into CNN architectures. Direct solve leads to memory overflow at very small output resolutions ($24^3$). Above computation benchmarks performed on a 2.4 GHz CPU chip.

In this paper we use the GauGAN architecture for the task of superresolution of turbulent fluid flow fields (See Fig. 1).

**Problem setup**    The main target application of this study is the super-resolution of turbulent flow fields. Fully resolving turbulence requires direct numerical simulation (DNS) that can resolve the smallest scales of the flow (Komogorov scale), which is prohibitively expensive. Therefore, the motivation of this study is to produce flow fields and flow statistics comparable to DNS at the cost of a low-resolution proxy, whilst strictly enforcing PDE constraints. To this end, we leverage high-resolution DNS data to train a deep neural network to learn the mapping between the low-resolution flow and its high-resolution counterpart. We compare between several algorithms for the task: the conventional trilinear interpolation which is not learning based, and various deep learning methods leveraging the GauGAN architecture for conditional generative modeling. We benchmark our PDEL in-the-loop hard constraining method against unconstrained training (denoted by "none" in figures and tables) and soft constrained training (denoted by "soft"), with and without hard constraining spectral projection at test time. The goal is to satisfy the imposed constraint and to evaluate the accuracy of the predicted flow fields using key domain-specific metrics.

**Dataset Description**    We use the Forced Isotropic Turbulence dataset from the Johns Hopkins Turbulence Database for this experiment **?**. The dataset consists of DNS at $1024^3$ resolution performed by solving the Navier-Stokes equations using the pseudo-spectral method. The dataset consists of 5028 frames (time steps) of data, each with 3 velocity components. For this experiment, we use all simulation frames starting from the 16th time step, since the initial frames consist of underdeveloped flow. Furthermore, since $1024^3$ resolution is practically impossible to fit into modern GPUs, we use a subsampled version of the data at $128^3$ as high-resolution targets, and further subsampled $32^3$ fields as low-resolution inputs. Subsampling for the high-resolution field is performed by uniformly sampling the original flow field at intervals of 8 in each dimension. For test and training splits, we take a random subset of $70/10/20\%$ of the original data as training, validation and test splits.

**Evaluation metrics**    Since the superresolution task constraining on the low-resolution input is mathematically underdetermined, it is not possible to recover the exact velocity field. Further, given the chaotic nature of turbulence, a single low-resolution flow field corresponds to various different realizations of high-resolution flows. Therefore, we refrain from directly comparing the norm of the difference in velocity fields. Instead, we compare the distributions of various key flow statistics, as outlined by **?**, which are more informative from a turbulence modeling standpoint. In Tab. 1, we report the Kolmogorov-Smirnov (KS) statistic between the ground truth test set distributions and the distributions generated by various models conditioning on the low-resolution test inputs. We also report the mean difference between the distributions, i.e., difference between the mean of the modelled and ground truth distributions in units of ground truth distribution standard deviation.

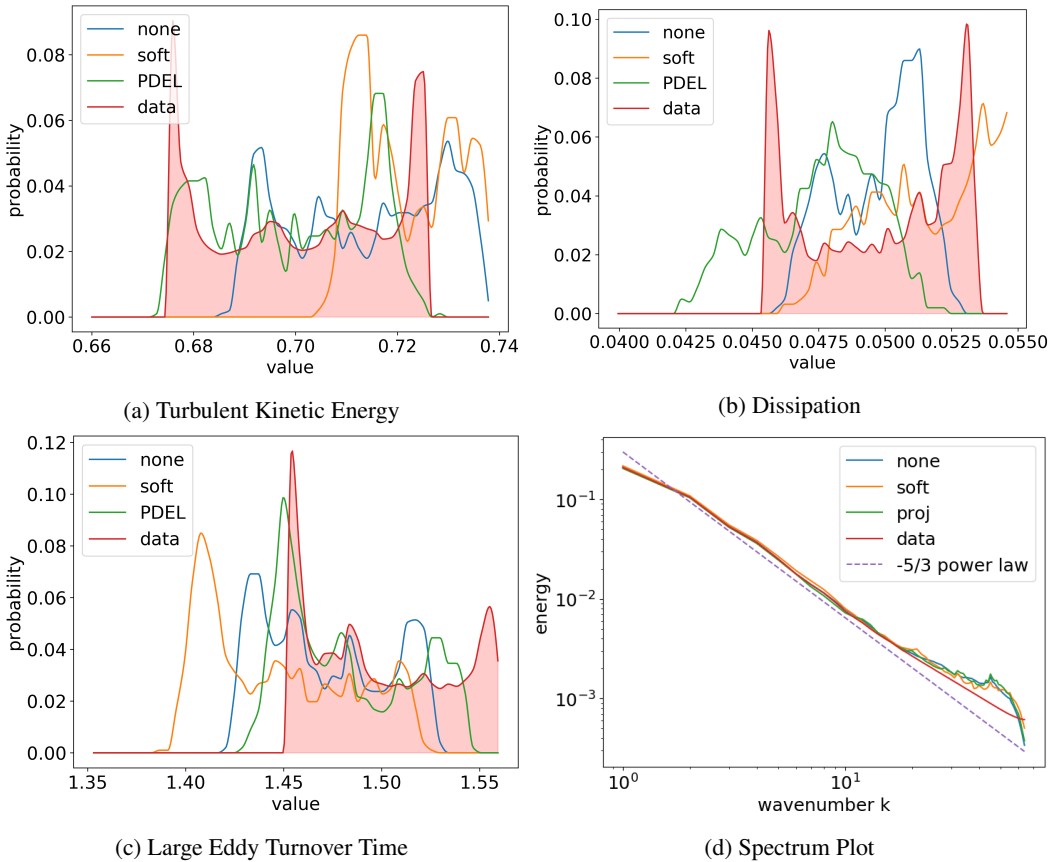

Figure 3: Probability Density Function plots of samples generated by various models as well as ground truth distributions (highlighted in red). The quality of the distributions compared against the ground truth distributions are captured well by the KS statistic and mean difference measurements in Tab. 1. (d) Shows the spectrum, where all methods trail the data distribution well in the low wavenumber regime and deviate at higher wavenumbers.

The flow statistics in Tab. 1 is defined as below. For simplicity, we denote the different velocity components using Einstein notation, and use angle brackets $<>$ to denote spatial averaging.

- Total kinetic energy, $E_{\text{tot}} = \frac{1}{2} < u_i u_i >$

- Dissipation, $\epsilon = 2\nu < \sigma_{ij}\sigma_{ij} >$, where $\sigma_{ij} := \frac{1}{2}\left(\frac{\partial u_i}{\partial x_j} + \frac{\partial u_j}{\partial x_i}\right)$, and $\nu = 0.000185$ is a constant for fluid viscosity.

- Large eddy turnover time: $T_L = L/u'$, where $L = \frac{\pi}{2u'^2} \int \frac{E(k)}{k} dk$ and $u' = \sqrt{(2/3)E_{\text{tot}}}$

**Results** The main quantitative results for this experiment are presented in Tab. 1, whereas a visualization of the distributions regarding various key flow statistics are presented in Fig. 3. We observe from empirical evaluations that training with the hard constraining layer in-the-loop effectively imposes the solenoidal constraints (zero residue), and enforcing the hard constraint at training time achieves more accurate flow field distributions measured by various key flow statistics. We note that although this method is not the most accurate for the dissipation statistic, presumably because of discrepancies in the high wavenumber regime (where dissipation occurs), the overall mean statistics and individual statistics for the other metrics are superior compared to all the other methods.

| Constraint Type | | No PDE Layer | | | PDEL at test time | | | In-the-loop |
| --- | --- | --- | --- | --- | --- | --- | --- | --- |
| | | Trilinear | None | Soft | Trilinear | None | Soft | PDEL |
| Residue($\downarrow$) | | 3.597 | 19.763 | 0.150 | **0.000** | **0.000** | **0.000** | **0.000** |
| KS Stats ($\downarrow$) | tkenergy | 1.000 | 0.308 | 0.712 | 1.000 | 0.216 | 0.632 | 0.163 |
| | dissipation | 1.000 | 0.283 | 0.549 | 1.000 | 1.000 | 0.332 | 0.422 |
| | eddytime | 1.000 | 0.388 | 0.599 | 1.000 | 0.229 | 0.487 | 0.276 |
| | mean | 1.000 | 0.326 | 0.620 | 1.000 | 0.482 | 0.484 | **0.287** |
| Mean Diff ($\downarrow$) | tkenergy | 6.227 | 0.745 | 2.192 | 6.593 | 0.396 | 1.845 | 0.106 |
| | dissipation | 16.245 | 0.016 | 1.301 | 16.732 | 2.690 | 0.731 | 0.804 |
| | eddytime | 9.343 | 0.878 | 1.516 | 10.037 | 0.436 | 1.125 | 0.591 |
| | mean | 10.605 | 0.546 | 1.670 | 11.121 | 1.174 | 1.234 | **0.500** |

Table 1: Comparison of generated distributions on test set against ground truth distribution on test set. Smaller values indicate a better match between distributions. Results indicate that while soft constraining can encourage the network to adhere to constraints, its residue is nonzero, implying that the imposed constraint is not strictly satisfied. Spectral based projection method can effectively eliminate residue. Training with the PDE layer in-the-loop eliminates residue and achieves greater accuracy on key statistical quantities, as compared to unconstrained and soft constrained cases, even when hard constraints are enforced at test time.

## 6 CONCLUSIONS AND FUTURE WORK

Enforcing hard physical constraints to solutions generated using neural networks are essential for their application to important scientific problems. In this paper, we propose a novel differentiable spectral projection layer for neural networks that efficiently enforces spatial PDE constraints using spectral methods, yet is fully differentiable, allowing for its use as a layer in neural networks that supports end-to-end training. Further, we show that its computational cost is cheaper than a regular convolution layer. We demonstrate its use in an important class of physics problems – incompressible turbulent flows, where the divergence-free PDE constraint is required. We are able to train a 3D Conditional Generative Adversarial Network (CGAN) for turbulent flow superresolution efficiently, whilst guaranteeing the spatial PDE constraint of zero divergence. Further, our results show that the model produces more accurate flow statistics when trained with hard constraints imposed via the proposed novel differentiable spectral projection layer, as compared to soft constrained and unconstrained counterparts.

Some key limitations of this work are: (i) the method is applicable in its current form only to flows with periodic boundary conditions; (ii) we only develop a method for linear spatial constraints and (iii) we only consider statistically steady flows. In future we will address all the above limitations to extend our work to more general sets of nonlinear unsteady constraints with arbitrary boundary conditions.

## APPENDIX

## A   MATHEMATICAL DERIVATION FOR SPECTRAL PROJECTION

The solution to the Lagrangian multiplier method for enforcing the solenoidal conditions involves inverting the left-hand-side matrix in Eqn. 7. Since $I, A, 0$ are block matrices, the inverse can be represened by

$$
\begin{bmatrix} I & A^T \\ A & 0 \end{bmatrix}^{-1} = \begin{bmatrix} I - A^T(AA^T)^{-1}A & A^T(AA^T)^{-1} \\ (AA^T)^{-1}A & -(AA^T)^{-1} \end{bmatrix} \tag{14}
$$

Hence the projected vector in spectral space can be computed as:

$$
\hat{\boldsymbol{F}} = P\boldsymbol{F} + Q\boldsymbol{B} \tag{15}
$$

The second term in the equation above drops out since $\boldsymbol{b} = 0$ for solenoidal constraints. More specifically for spectral methods, the matrix $A$ can be represented as three diagonal matrices for the wavenumbers in the three dimensions multiplied by the imaginary number $i$:

$$
A = \begin{bmatrix} \ddots & & \ddots & & \ddots & \\ & A_1 & & A_2 & & A_3 \\ & \ddots & & \ddots & & \ddots \end{bmatrix} \tag{16}
$$

$$
= [A_1 \quad A_2 \quad A_3] \tag{17}
$$

The only matrix inverse associated is $AA^T$, whose value can be computed by block matrix multiplication:

$$
AA^T = A_1^2 + A_2^2 + A_3^2 \tag{18}
$$

Given that $A_1, A_2, A_3$ are diagonal matrices (eliminating the terms regarding the $[0, 0, 0]$ mode), its inverse can be computed by directly inverting the diagonal terms:

$$
(AA^T)^{-1} = \frac{1}{A_1^2 + A_2^2 + A_3^2} \tag{19}
$$

Hence the linear projection matrix can be written as:

$$
I - A^T(AA^T)^{-1}A = I - \frac{1}{A_1^2 + A_2^2 + A_3^2} \begin{bmatrix} A_1^2 & A_1 A_2 & A_1 A_3 \\ A_1 A_2 & A_2^2 & A_2 A_3 \\ A_1 A_3 & A_2 A_3 & A_3^2 \end{bmatrix} \tag{20}
$$

$$
A^T(AA^T)^{-1} = -\frac{1}{\sum_{j=0}^{3} A_j^2} \begin{bmatrix} A_1 \\ A_2 \\ A_3 \end{bmatrix} \tag{21}
$$

Recovering the same solution as in Eqn. 13. More specifically for the divergence-free condition, we have:

$$
A_j = diag(-i\boldsymbol{k}_j) \tag{22}
$$

$$
\boldsymbol{B} = \boldsymbol{0} \tag{23}
$$

Hence the spectral projection step can be further simplified as:

$$
\hat{\boldsymbol{F}} = \boldsymbol{F} - \frac{\boldsymbol{k} \cdot \boldsymbol{F}}{\boldsymbol{k} \cdot \boldsymbol{k}} \boldsymbol{k} \tag{24}
$$

It is easy to show that the result is divergence-free, since:

$$
-i\boldsymbol{k}\hat{\boldsymbol{F}} = -i\boldsymbol{k}\boldsymbol{F} + i\boldsymbol{k}\boldsymbol{F} = 0 \tag{25}
$$

It is also easy to show that the projection is orthogonal to the solution space, since the dot product between the $\hat{\boldsymbol{F}} - \boldsymbol{F}$ and $\hat{\boldsymbol{F}}$ is zero:

$$
(\hat{\boldsymbol{F}} - \boldsymbol{F}) \cdot \hat{\boldsymbol{F}} = -(\frac{\boldsymbol{k} \cdot \boldsymbol{F}}{\boldsymbol{k} \cdot \boldsymbol{k}} \boldsymbol{k}) \cdot (\boldsymbol{F} - \frac{\boldsymbol{k} \cdot \boldsymbol{F}}{\boldsymbol{k} \cdot \boldsymbol{k}} \boldsymbol{k}) \tag{26}
$$

$$
= \boldsymbol{0} \tag{27}
$$

| Notation | Description |
|---|---|
| Conv($n$) | 3D convolution block with $n$ output layers |
| SResBlock | SPADE Residue Block consisting of
SPADE + ReLU + Conv + SPADE + ReLU + Conv |
| Up($n$) | Upsampling layer with nearest neighbor interpolation with scaling factor of $n$ |
| IN | 3D Instance Normalization Layer |

Table 2: Description of module notations in Fig. 1

## B    MODEL AND TRAINING DETAILS

We use the GauGAN architecture (with schematics as shown in Fig. 1) for the conditional flow field generation task. The abbreviated names for the various modules are given in Tab. 2. Our model differs from the original GauGAN model in two distinct aspects. First, our architecture utilizes 3 dimensional convolutions instead of the 2 dimensional counterparts in the original GauGAN architecture. Second, for our hard constrained case, we append the spectral projection layer to the end of the architecture for enforcing hard constraints.

For training the model, we use multiresolution discriminator loss as in **?** across 3 discriminators. We train the model with batch size of 18 (across 6 Volta V100 GPUs) with the Adam optimizer using a learning rate of $2E - 4$. The soft constrained model uses a residue penalty factor of $0.01$.

## C    ADDITIONAL VISUALIZATION

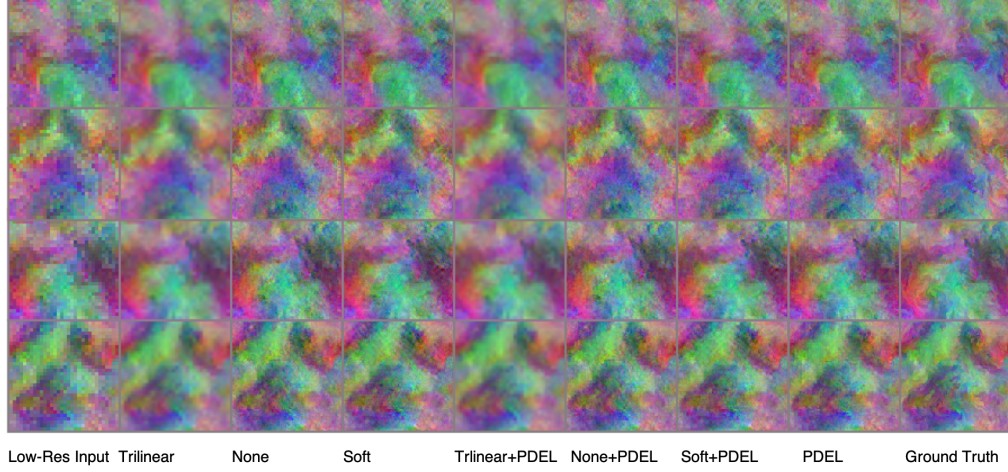

Low-Res Input    Trilinear        None        Soft        Trlinear+PDEL  None+PDEL    Soft+PDEL      PDEL         Ground Truth

Figure 4: Visualizations for low resolution inputs from the test set, predictions by various models, as well as ground truth flow fields. The flow fields are colored by mapping the three velocity components to RGB channels respectively. The 2d images are slice plots in the $z$ dimension.

