# OpenReview forum: "Enforcing Physical Constraints in Neural Neural Networks through Differentiable PDE Layer"
_ICLR.cc/2020/Conference — Reject_

### Official Review · AnonReviewer3 · 2019-10-16
**Official Blind Review #3**

**Rating:** 6

**Review:**

The paper describes a way to efficiently enforce physical constraints expressed by linear PDEs on the output of a neural network. The idea is to have, as a last layer of the network, a projection onto the constrained solution space, and to back-propagate through it. That projection layer is made efficient for high-dimensional outputs via the fast Fourier transform (FFT), exploiting a well-known numerical trick. Importantly, the proposed strategy is very general, and can indeed be used with any PDE constraint that is a linear combination of differential operators.

First, I would like to mention to the AC that the authors apparently forgot to run bibtex. The submitted paper had no bibliography and all references are "?". So technically the original submission was incomplete and would probably have to be rejected, on the grounds that it was impossible to check whether the literature references are appropriate. The authors have rectified this through a comment pointing to an (anonymous) version with references. For me, this is ok.

The research direction of the paper is a hot topic: how to reconcile data-driven deep learning with analytic physical models is, arguably, one of the big research questions holding back the wide-spread use of deep networks in several natural sciences (e.g., environmental and climate science, hydrology, etc.). Work in this direction could have a significant impact, and the approach developed the paper is rather general and, as far as I can tell, correct.

The experiments use the rather challenging task of super-resolving turbulent flow, subject  to the Navier-Stokes constraints. Experiments are run on synthetic data from the JHTD simulation dataset. This is hard to avoid, since dense ground truth for flow fields is impossible to obtain, nevertheless it  would have been more convincing to also show at least qualitative results for a real flow dataset. It is also not completely satisfactory that the experiments must use even as OUTput a significantly downscaled version of the dataset with a resolution that would not be tremendously useful in practice - this implicitly acknowledges that the paper, while reducing the computational cost compared to the direct projection method, did actually not overcome the most critical bottleneck one faces when combining high-dimensional physics simulation with deep learning: namely, that it is not tractable to simply store physical fields as explicit 3D/4D voxel grids if one wants to work with them on the GPU.

Another slightly bothering choice in the experiments is to compare only distributions. While I see the point that the prediction is, by its nature, ambiguous; I still think one should look at both flow statistics and at the actual flow field difference. Ambiguity is not a very convincing justification to not even try to predict correctly - by the author's definition any super-resolution task is ambiguous, still one aims, for instance, to recover the correct image, not just a plausible one. In a sense it is the whole point of prior knowledge (which the PDEs are in a learning context) to bring the solution closer to the right answer, when the data alone cannot do the job. In practice, a prediction that is very close to the true flow field, but ever so slightly violates the physical constraints is often more useful than one that strictly satisfies the constraints, but is way off.

One comment on the presentation: I feel that the discussion of soft constraints could be more extensive and more balanced. I agree that they are less principled, they do not devalue the present work. But we know, both for variational methods and for pre-deep learning methods, that soft constraints work rather well in practice, especially with an adaptive weight that gradually tightens the constraints. So it would be in order to not just dismiss the alternative in a half-sentence, but to give it proper consideration - especially since in terms of dissipation, it actually performs better than the hard constraint in the experiments.

Overall, I find the topic important and the presented work is a sensible and nicely generic step forward. On the negative side, the paper does not fully deliver on the promise to make physics constraints in deep networks usable in practice. My rating reflects my impression that the bibliography and references are probably correct - this should be checked before reaching a final decision.


**Experience Assessment:**

I have read many papers in this area.

**Review Assessment: Checking Correctness Of Derivations And Theory:**

I assessed the sensibility of the derivations and theory.

**Review Assessment: Checking Correctness Of Experiments:**

I assessed the sensibility of the experiments.

**Review Assessment: Thoroughness In Paper Reading:**

I read the paper at least twice and used my best judgement in assessing the paper.

---

### Official Review · AnonReviewer2 · 2019-10-22
**Official Blind Review #2**

**Rating:** 3

**Review:**

This paper proposes to use a differentiable FFT layer to enforce hard constraints for results generated by a CNN. This is demonstrated and evaluated for a 3D turbulence data set (an interesting and challenging problem), and evaluated for a single case.

While this goal is good by itself, and the domain of applications is a very interesting one, the paper gives the impression of being preliminary, and the claims for the proposed constraints are a bit too generic, in my opinion.

First, the FFT effectively only yields a somewhat specialized method for projection onto the set of admissible solutions, and is demonstrated only for a single constraint, i.e., to make the flow field solenoidal. The same goal can actually be reached in different ways, e.g., by inferring a vector potential as proposed by Kim et al. 2019 in the "DeepFluids" paper. The latter employs a curl formulation, and as such is less general, but probably faster than the FFT based method proposed here.

In addition, the paper unfortunately contains only a single example. Here, several variants (no constraint, soft constraint, and the proposed method) are evaluated in addition to simpler interpolation methods. Visually, I could not really make out differences in figure 4. The metrics in table 1 look interesting, although it, e.g., didn't get clear to me what the "KS stats" mean. The graphs in figure 3 also paint a somewhat varied picture. While some regions seem to be well represented, others are clearly there in the references, but missing in one of the inferred versions.

I was wondering in general - what is the intuition for divergence-freeness improving the TKE, for example? It's neat to see the metrics improve, but wouldn't one expect that a projection onto divergence free flows rather removes energy from the solutions, and hence maybe yield values that are too low?

I think the paper could be improved by first evaluating the method for a series of smaller two-dimensional examples, before tackling a full 3D flow. This would simplify comparisons to other methods, and help to illustrate the properties of the method. Ideally, other constraints than enforcing divergence-freeness could be demonstrated to show the generality of employing an FFT projection in the loss function. So currently, I think this paper is not quite ready for a conference such as ICLR. It would be important to demonstrate that the result shown here is not an "outlier", but that the improvements are a general trend obtained via the proposed method.

**Experience Assessment:**

I have published in this field for several years.

**Review Assessment: Checking Correctness Of Derivations And Theory:**

I carefully checked the derivations and theory.

**Review Assessment: Checking Correctness Of Experiments:**

I carefully checked the experiments.

**Review Assessment: Thoroughness In Paper Reading:**

I read the paper thoroughly.

---

### Official Review · AnonReviewer1 · 2019-10-23
**Official Blind Review #1**

**Rating:** 3

**Review:**


This work develops a differentiable spectral projection layer to enforce spatial PDE constraints using spectral methods, to achieve the introduction of the physical constraints in the end-to-end network without damaging the intrinsic property of the network. Analysis of computational cost shows the proposed layer is cheaper than the convolutional layer. The experimental comparison demonstrates the superiority of the proposed method. In my viewpoint, the novelty of this paper is somewhat novel.

This paper focuses on designing the PDE layer to constrain the network output without additional loss. Some constraint sets for comparison are clearly performed and then authors present the proposed spectral projection layer. From the mechanism of solving, the FFT (IFFT) operator is important component in this layer. It is curious about the role and importance between FFT (IFFT) and spectral projection. If possible, the authors maybe provide some analyses of these two components to deeply recognize the proposed layer.

A series of compared experiments are conducted to verify the effectiveness of the proposed PDEL. But it is a little confusing in Table 1, e.g., why the second column obtains the second better in the all mean values. I know its result has a high score of residue. It will be better if authors can clarify their causes and analysis clearly.


**Experience Assessment:**

I have published in this field for several years.

**Review Assessment: Checking Correctness Of Derivations And Theory:**

I carefully checked the derivations and theory.

**Review Assessment: Checking Correctness Of Experiments:**

I carefully checked the experiments.

**Review Assessment: Thoroughness In Paper Reading:**

I read the paper at least twice and used my best judgement in assessing the paper.

---

### Decision · Program_Chairs · 2019-12-19

**Decision:**

Reject

**Comment:**

This paper introduces an FFT-based loss function to enforce physical constraints in a CNN-based PDE solver.  The proposed idea seems sensible, but the reviewers agreed that not enough attention was paid to baseline alternatives, and that a single example problem was not enough to understand the pros and cons of this method.